# Management of Non-Communicable Diseases in Kosovo: A Scoping Review

**DOI:** 10.3390/ijerph20043299

**Published:** 2023-02-13

**Authors:** Ilir Hoxha, Valid Apuk, Besfort Kryeziu, Premtim Rashiti, Mrika Aliu, Alejandro Gonzalez Aquines, Olga Khan, Ha Thi Hong Nguyen

**Affiliations:** 1The Dartmouth Institute for Health Policy and Clinical Practice, Geisel School of Medicine at Dartmouth, Lebanon, NH 03766, USA; 2Evidence Synthesis Group, 10000 Prishtina, Kosovo; 3Research Unit, Heimerer College, 10000 Prishtina, Kosovo; 4Swiss Tropical Public Health Institute, 4000 Basel, Switzerland; 5National Institute of Public Health of Kosovo, 10000 Prishtina, Kosovo; 6Emergency Clinic, University Clinical Center of Kosovo, 10000 Prishtina, Kosovo; 7The World Bank, 10000 Prishtina, Kosovo; 8Faculty of Health Studies, University of Bradford, Bradford BD7 1DP, UK; 9The World Bank, 1020 Vienna, Austria

**Keywords:** NCD management, diabetes, hypertension, breast cancer, cervical cancer

## Abstract

Background—Non-communicable diseases (NCDs) affect a growing share of the population in Kosovo. The country faces challenges with NCDs management, specifically detecting, screening, and treating people with NCDs. Objective—To assess the management of NCDs, including the inputs that influence the provision of NCDs and outcomes of NCD management. Eligibility criteria—Studies had to report NCD management in Kosovo. Sources of evidence—We systematically searched Google Scholar, PubMed, Scopus, and Web of Science. Charting methods—The data were charted by two researchers. We extracted data on general study details and design and information on the management and outcomes of NCDs in Kosovo. Synthesis of results—For the mix of studies that were included in the review, thematic narrative synthesis was used. We developed a conceptual framework based on health production core components to analyze the data. Results—Kosovo’s health care system is available to provide basic care for patients with NCDs. However, there are serious limitations in the availability of key inputs providing care, i.e., funding, medicines, supplies, and medical staff. Additionally, in terms of the management of NCDs, there are areas for improvement, such as limited application of clinical pathways and guidelines and issues with referrals of patients among levels and sectors of care. Finally, it is worth noting that there is overall limited information on NCD management and outcomes. Conclusions—Kosovo provides only basic services and treatment of NCDs. The data reporting the existing situation on NCD management are limited. The inputs from this review are helpful for existing policy efforts by the government aimed to enhance NCD care in Kosovo. Funding—This study is part of the research done for a World Bank review of the state of NCDs in Kosovo and was funded through the Access Accelerated Trust Fund (P170638).

## 1. Introduction

Kosovo is an upper-middle-income country with a population of 1.8 million, in South-Eastern Europe [1,2,3]. The average life expectancy in Kosovo is 71.9 [1] to 76.7 years [2,3]. The mortality rate of all causes has continued to fall in Kosovo, as with the rest of the European Region, though from 2006 to 2011, it has shown a slight trend of increase [3]. Non-communicable diseases (NCDs) affect a large share of the population in the country. About 21.6% of adults (18+ years) reported having a chronic disease in 2017 [1]. Women are more likely to report chronic diseases than men [1]. Cardiovascular diseases are the leading cause of morbidity and mortality, followed by malignant diseases and respiratory diseases [3,4,5,6]. Diabetes is also prevailing among the population [7]. Marginalized communities, such as Roma, Ashkali, and Egyptian people, are particularly vulnerable [8,9,10].

Due to convenient demographic trends, where most of Kosovo’s population is younger, there were few concerns about NCDs’ impact on the population’s health. However, demographic trends are changing with very swift changes in population lifestyle trends and exposure to environmental hazards, notably pollution. This all hints that Kosovo is, and will continue to be, facing an increasing number of people with NCDs and should adjust healthcare services provision to match these evolving needs.

NCD care in Kosovo is provided by all three levels of care, i.e., primary, secondary, and tertiary [10]. The primary healthcare system is the network of family medicine facilities. Secondary healthcare represents regional hospitals where specialized care is provided. Tertiary healthcare refers to research and teaching institutions that are supposed to provide high-end diagnosis and treatment. NCD care, as well as any type of care, is affected by numerous shortcomings of the health care system. For example, Kosovo has among the lowest ratios of physicians and nurses in Europe [11], with 1.44 physicians per 1000 inhabitants compared to the European Union (EU) average of 3.4 doctors per 1000 population. The relatively low and stagnant level of salaries in the public sector, the limited capacity of absorption of additional staff in the health care system, and the difficult working conditions, especially for the nurses, explain the concerning trend of migration of health professionals to European Union countries. Furthermore, the public health spending in Kosovo as a share of GDP is low relative to the regional and global per capita income comparators. In 2015, public health expenditure in Kosovo was 2.9 percent of the gross domestic product (GDP) and had been at such levels over the last two decades [4,12,13,14]. This is considerably lower than the EU average or averages for countries in South-Eastern Europe [4]. Private health expenditure, almost entirely in the form of out-of-pocket payments by households at the point of service, contributes an estimated 40 percent of total spending in the sector [10,11]. The application of standards of care is limited [15]. Medical practice is left to physician education, experience, and willingness to perform clinical procedures. This impacts how care is provided for any type of patient, including patients with NCDs.

Management of NCDs is complex and resource consuming undertaking, which includes detecting, and treating these diseases. The evidence on the management of NCDs in Kosovo is scattered in different reports and peer review publications. There is no common picture and understanding of the current situation regarding the management of NCDs in the country. Several serious efforts to improve the situation are under discussion, including developing a country NCDs strategy. To support the government and other interested stakeholders, we decided to perform a scoping review that will help generate a clearer picture of NCDs management in Kosovo with the hope that it will support decision making at times when there are ambitions to put forward strategies and policies that will address the current situation with NCDs in the country. Henceforth, the objective of this study is to assess the inputs that influence the delivery of NCDs care, the process of management of NCDs, and outcomes from such management of NCDs.

## 2. Methods

We performed a scoping review. A scoping review can provide an overview of the available research evidence [16,17,18,19]. Therefore, we performed a scoping review of the grey literature and peer review publications in line with the PRISMA extension for scoping reviews (PRISMA-ScR) statement [16].

### 2.1. Protocol and Registration

The protocol was developed for internal purposes, stating all eligibility criteria, search databases, search terms, data extraction sheet, and frameworks for analysis. However, the protocol was not published in advance.

### 2.2. Information Sources, Search Strategy, and Eligibility Criteria

We systematically searched four databases, i.e., Google Scholar, PubMed, Scopus, and Web of Science, from the beginning to January 2023, when the search was last updated. We performed two searches in Google Scholar. One was performed with Albanian, and another one with English search terms. The search strategy in the databases (i.e., PubMed) targeted relevant peer-review papers. The search strategy in Google Scholar targeted peer review papers and the grey literature, i.e., publications and reports by different agencies and institutions (national and international). The search strategy consisted of terms related to NCDs and terms referring to the country Kosovo (Appendix A). Specifically, we used terms related to specific NCDs (i.e., hypertension, diabetes, breast cancer, cervical cancer) or general terms for NCDs (i.e., non communicable diseases, NCDs, etc.) and, with the exception of the Albanian language search in Google Scholar, search terms were related to the country, i.e., Kosovo. For the Albanian language search, we added the search term “chronic diseases”, as NCDs are often noted with such a term. We omit using country in Google Scholar search for the Albanian language to keep the search more comprehensive. 

Studies were included in case they reported valid information on NCDs management in Kosovo. Key conditions (NCDs) of interest were diabetes, hypertension, breast cancer, and cervical cancer. Nevertheless, there was interest in NCDs in general. Therefore, if papers reported information on other NCDs or, in general, for NCDs, they were included. There were no restriction criteria on publication time or publishing language or type of study design. Studies were excluded only to remove duplicate data.

### 2.3. Selection of Sources of Evidence and Data Extraction

The acceptability of studies for inclusion was determined by screening titles and abstracts, followed by a full-text evaluation by an independent reviewer. Three researchers (IH, VA, and BK) examined the search results, analyzing titles and abstracts before conducting a full-text review. To categorize and extract the data, a data extraction sheet was created. The team gathered data independently and in duplicate for the study’s general details and information on NCDs management. We have not contacted authors or made any other effort to gather additional data related to included papers. We have used only information from included publications. When it comes to NCDs management, the extraction sheet had three main sections for collecting data, which largely follow the conceptual framework used for data analysis (Figure 1). The first section collected information management of NCDs (the process), i.e., diagnosis use in treatment (i.e., staging, risk assessment), counseling services, treatment of NCDs, clinical pathways, interaction with patients, referral of patients among levels of care, referral of patients to the private sector, application of protocols, and any other relevant info that did not fit in previous categories. The second section collected data on the factors (inputs) influencing the management of NCDs, i.e., funding of the healthcare system, supply with medication and medical supplies, supply and capacity of healthcare staff, availability of clinical guidelines, and other factors that would not fit in other categories. Finally, the last section collected information on outcomes of management of NCDs (outcomes), i.e., satisfaction with care, access to care, NCD-related mortality rate, NCD-related hospitalization rate, and NCD-related burden of disease. All section data were collected for specific NCDs, i.e., diabetes, hypertension, breast cancer, cervical cancer, or any NCD. 

### 2.4. Critical Appraisal of Evidence

There was no standard quality assessment of studies. However, the team double-checked that the sources used to publish the findings were reliable. This is related to the fact that we wanted to gather all credible evidence to provide a thorough scoping review exercise that will benefit all parties interested in the situation with NCDs in Kosovo, be that the government or other national or international actors. 

### 2.5. Synthesis of Results

The retrieved data were subjected to thematic synthesis [20]. The mix of studies considered in the review and the conceptual framework for data analysis lends to such a strategy. Thematic synthesis was carried out by three researchers (IH, VA, and PR). Each researcher independently performed the thematic analysis. Findings from all studies were collated under main themes and subthemes following a conceptual framework (Figure 1) for analysis designed using a health services production framework with three main elements, i.e., inputs, processes, and outcomes. Hence, our main themes were the management of NCDs (processes), inputs that determine the management of NCDs (inputs), and finally, outcomes of management of the NCDs (outcomes). Different subthemes were determined based on our interest and relevance of such information in understanding the NCDs management in the country, i.e., availability of protocols, referral of patients, etc.

## 3. Results

### 3.1. Selection of Sources of Evidence

We identified 950 documents across four databases, i.e., 500 in Google Scholar, 172 in PubMed, 155 in Scopus, and 123 in Web of Science (Figure 2). One hundred and thirty-five documents were found with a manual search. One hundred and twenty-four duplicate documents were excluded from screening. Nine hundred and sixty-one documents were screened for eligibility by looking at the title and abstracts. Four hundred and three documents were reviewed in full text to assess eligibility. Finally, 65 documents were included in the evidence synthesis [21,22,23,24,25,26,27,28,29,30,31,32,33,34,35,36,37,38,39,40,41,42,43,44,45,46,47,48,49,50,51,52,53,54,55,56,57,58,59,60,61,62,63,64,65,66,67,68,69,70,71,72,73,74,75,76,77,78,79,80,81,82,83,84,85].

### 3.2. Characteristics of Sources of Evidence

We present the document characteristics in Table 1 and the Appendix A. The documents were published between 2002 to 2022. Most of the documents, i.e., 57, were published from 2010 and on. Eighteen documents were published within the last five years. Most (i.e., 35) of the documents were peer reviewed papers published in academic journals, and 23 were different reports published by different agencies, and seven were policy briefs. Twenty-four documents reported information on inputs needed for NCDs management. Forty-one documents reported information on the processes related to NCDs management. Finally, 23 documents reported information on the outcomes of NCDs in Kosovo. Fourteen studies reported information on diabetes, 11 reported on hypertension, 13 documents reported information on breast cancer, five reported on cervical cancer, four reported on other NCDs, and 33 documents reported overall information on NCDs. Results of individual sources of evidence are presented in the Appendix A. In the sections below, we provide a summary of these results.

### 3.3. Inputs for NCDs Management

We found some evidence for medication and supplies, supply with healthcare staff, availability of clinical guidelines, and financing of NCD care. Kosovo’s healthcare system is underfunded, which is reflected in the lack of medical supplies and treatment within institutions [42]. The problems of supply with medication and medical supplies are a reoccurring issue for public health care providers at all levels of care [26,37,39,49,51]. This leaves the patients to pay for the drugs themselves [28]. In recent times, there have been significant improvements in the supply of drugs to health institutions [28].

Kosovo has one of the lowest supplies of healthcare staff in Europe. The immigration of health personnel from Kosovo has aggravated the supply of medical staff [43]. The capacities and clinical competence of medical staff are another domain where advancement is needed [36]. There have been several efforts by institutions to develop capacities for treating patients with NCDs. However, there is still a long way to go before improvements become dominant in the provision of care [42,48]. Supply issues hamper the provision of care related to NCDs [23,48]. The availability of clinical guidelines has been supported primarily by external assistance and has resulted in positive effects [31,59,84], but overall, it remains limited [59,76,84].

The current benefit package of care in Kosovo is financially unsustainable with public funding, resulting in frequent stock outs of basic materials and pharmaceuticals in health institutions. Although part of the benefit package, a large amount of pharmaceuticals are paid for out of pocket [24,41,49]. The establishment and functionalization of the Health Insurance Fund (HIF) has not been able to address funding gaps in the healthcare system regarding NCDs [65]. Another important problem are informal payments in the public sector [46].

### 3.4. Management of NCDs

We found some evidence on diagnostic, treatment, and counseling services. We also found some evidence for the application of clinical pathways and guidelines, as well as referral of patients with NCDs among levels of care and the private sector. Kosovo’s public health care system provides only basic diagnostics and treatment of NCDs. For hypertension diagnosis and assessment, healthcare providers use blood pressure measurement, blood tests, urine sample examination, and physical exams of patients [55]. Evidence reveals shortcomings in the assessment and examination of patients [61]. When it comes to breast or cervical cancer, unfortunately, a large proportion of cases are diagnosed in their late stages [48]. This is mainly due to ineffective breast and cervical cancer screening programs [48,55]. There is also a lack of information about disease staging [56]. Currently, several methods or tests are commonly used for breast cancer screening, including mammography, breast self-examination, clinical breast examination, and ultrasound [27,48,55].

Primary healthcare plays a vital role in the treatment of NCDs, i.e., diabetes and hypertension. However, there is a preference for specialist care [50]. Patients have access to general and specialized care in all three levels of care as well as the private sector. There is also evidence of the provision of comprehensive care [68,72]. However, as in the case of hypertension, it is often uncertain the extent of the management of patients via a holistic approach and therapeutic personalization [33]. In the case of breast cancer, there is widespread evidence of insufficient decision-making that is not based on best medical practice standards and inappropriate treatment and follow-up by providers and patients [23]. The surgical treatments for breast cancer have changed dramatically, from radical mastectomy to breast-conserving surgery [83]. About one-third of women with breast cancer in Kosovo still have a mastectomy [73]. Additionally, in the case of cervical cancer treatment, there are issues with the quality of the services, the coordination of care, and the establishment of a definitive diagnosis and treatment [48].

When it comes to counseling services, there is a lack of medical staff, especially nurses, who can counsel patients with NCDs [42]. If available, the consulting doctors are either family doctors or specialists in the public or private sectors [42]. Projects with external aid have supported primary healthcare institutions to establish motivational counseling services for type 2 diabetes and hypertension [71]. There is not much evidence on the availability of counseling services on breast cancer or other cancers. Earlier evidence points out that Kosovo has yet to develop a capacity to provide adequate counseling services in case of diagnosis and detection [54].

Work with clinical pathways and guidelines is primarily initiated and assisted through external aid [67,71,75]. Other than that, we found no evidence to support the implementation of clinical pathways in care delivery. The lack of a defined patient pathway for suspected cases of breast or cervical cancer exemplifies the problems related to NCDs management in the country [48]. From the initial presentation to definitive therapy, the patient’s care is challenging and ineffective. [23] Self-referral to secondary or tertiary clinics persists [23]. The application of protocols also seems to be limited according to existing evidence [33,48], but there are several important efforts to bring them to practice [48,74,84].

The existing referral mechanism for patients with NCDs is inefficient [61,77]. Regulations defining referral are not enforced [77]. Patients frequently forego primary care in favor of secondary or tertiary care, although primary care facilities can provide the required services [77]. The healthcare system needs guidelines that clearly define the duties and responsibilities of primary, secondary, and tertiary healthcare institutions, leaving little room for a formal referral system to be established [86]. The lack of a functional Health Information System (HIS) [37] and financial incentives via health financing arrangements [32] do not help with this situation. The establishment of Kosovo Hospital and University Clinical Services (KHUCS) did not improve referrals among institutions. [63]. Evidence suggests numerous issues with referrals in the private sector [26,28,32,51,57].

### 3.5. Outcomes from the Management of NCDs

We found little information on the outcomes of NCDs. We found some information on satisfaction with care, access to care, NCDs-related hospitalization rates, NCDs-related mortality rates, and little-to-no information on the NCDs-related burden of disease. Roughly half of the primary care users in Kosovo are satisfied with the overall medical experience [45]. Diabetic patients are most satisfied with visits to diabetes specialists and endocrinologists compared to primary care physicians [50]. Breast and cervical cancer patients appreciate care received at the secondary and tertiary care level [42].

Access to NCDs treatment can be limited for minority ethnic groups [44,55], except for the Serb population [54]. This is related to the difficulty of paying for medical treatment [53], availability of care [54], and discrimination [52]. Access is not limited only to ethnic groups. Access to care can be limited for those living in poverty, the elderly, people with disabilities, those living in remote areas, and women [44,54].

The number of diabetic and hypertensive hospitalizations in Kosovo has been increasing [74]. According to statistics, over half of the hospital deaths are caused by NCDs [29,30]. Arterial hypertension is one of the major risk factors for mortality and morbidity [68], often accompanied by atherosclerosis [31] or resulting in heart failure [34]. Cardiovascular, renal and lung illness, chronic back pain, and gastritis are the most common NCDs [40]. The epidemiological data suggest increasing breast and cervical cancer deaths. This is observed for other cancers, as well [60]. Women who live with cancer, without even essential medical or social assistance or support, bear the hidden weight of the disease [23].

## 4. Discussion

### 4.1. Summary of Findings

Kosovo’s health care system can provide basic care for patients with NCDs. At times, advanced care is available. However, there are serious limitations in the availability of key inputs for the provision of care, i.e., funding, medicines, supplies, as well as the medical staff. Additionally, in terms of the management of NCDs, there are shortcomings, such as limited application of clinical pathways and guidelines and issues with referrals of patients among levels and sectors of care. Finally, it is worth noting that there is overall limited information on NCDs management and, in particular, on NCDs outcomes.

### 4.2. Strengths and Weaknesses

A thorough and systematic search of the existing literature in some of the key databases, screening and extraction of data performed by at least two reviewers, as well as organized and conceptualized extraction and charting of data from studies, are some of the key strengths of this scoping review. Despite a thorough search, we may have missed relevant studies due to the general nature of our search, or documents may not have been registered in databases we have used. The depth of review in terms of understanding NCDs management may also be considered limited, which is largely conditional on the availability of data in the included papers. Some information may also be outdated. Nevertheless, it is worth noting that we found way more studies and information than we initially expected.

### 4.3. Context

To our knowledge, this is the first review of existing evidence on NCDs in Kosovo. It confirms many publicly known facts about the management of NCDs that are valid for most of the healthcare provided in the country. For example, studies have reported that the system is underfunded, [4,11,87] care availability is limited in quality and quantity, [4,83] issues exist relating to availability and impact of the application of clinical standards [4,15,84], and there are issues with supply with the workforce [11,88]. This review highlights the issues with a defragmented system of provision of NCD care, as well as efforts in the function of improvement of the existing situation. It does so by substantiating existing knowledge with the most recent available evidence. According to estimates, the worldwide economic burden of NCDs is anticipated to expand, and low- and middle-income countries will bear a greater portion of it as a result of population growth, aging, and globalization [89,90]. This will be a vital point to have in mind for policymakers in Kosovo, especially in light of changing demographic trends in Kosovo.

### 4.4. Implications for Policy, Practice, and Research

From this review, we can derive several learning points, i.e., issues to be addressed that can serve as the basis for the future work of relevant actors and, in particular, policymakers. Three main issues require the attention of policymakers and healthcare providers concerning the management of NCDs. Government and health care institutions (1) should put serious effort into improving the quality of care with regards to NCDs, (2) they should invest in human resources involved in the provision of care for NCDs, and (3) use health financing reform to protect the patients with NCDs financially with particular emphasis on vulnerable groups, such as minority communities, the elderly, and people from rural zones.

When it comes to quality improvement efforts, successful long-term treatment and management of NCDs in Kosovo will highly depend on the further transfer of knowledge, training of staff, and adoption, development, and implementation of clinical pathways and protocols. It will not be enough only to draw or adopt new protocols and standards. Also, external assistance can support but not replace internal efforts. Efforts will have to go deep into the roots of medical education, continuous medical education, and medical practice change. Radical efforts are needed in the adoption and implementation of clinical guidelines, as well as building the capacity and control mechanisms to ensure the clinical standards are enacted in NCDs care. Improvement efforts should touch all levels and sectors of care.

The Ministry of Health and healthcare institutions should engage in strategic planning that will mitigate the lack of clinical staff, especially medical doctors, and ensure long-term solutions for the proper supply of institutions with healthcare staff. The under-discussion NCDs strategy and medical staff payment level policies are perfect examples of that. In addition, incentives and measures are needed to be coupled with hospital development measures in the form of specialization of hospitals or the development of clinical excellence for the treatment of particular NCDs. This exercise is not just about proper supply numbers but also investment in the quality of their education and clinical excellence.

It is critical to construct health financial arrangements within existing health financing reform that ensure proper protection of people with NCDs, particularly from marginalized groups. Such mechanisms should ensure the availability of medications and services for people with NCDs. This is a long-term bet, but it is the only way to ensure sustainable protection of patients and their families from financial burdens associated with the healthcare they need. The under revision law on financing of healthcare and other policy measures that will follow are the moments for making such interventions.

This review notes the increasing amount of publication in relation to NCD care, as well as a trend in the use of advanced research designs to examine NCD care in Kosovo. Use of conjoint design in understanding clinical decision making with regard to NCDs care [84] or qualitative designs to understand the effects of NCDs-related counseling services on users of care are among notable examples. [81] These efforts should continue, be institutionally supported, and grow together with the data availability of NCDs care. Healthcare institutions are at a stage where they need specific information to understand existing problems and draw strategies that will address them.

## 5. Conclusions

Kosovo’s healthcare system is able to provide basic care for patients with NCDs. However, there are serious limitations in the availability of key inputs for the provision of care and proper management of NCDs that result in unfavorable NCDs outcomes. This review provides updated and reliable information that could be useful in the design of policy measures that aim to improve NCD management in Kosovo. Limited information on NCDs management, which is confirmed in this review, should be a call for a serious effort to document the management of NCDs in the country. Improvements will be easier to track, document, and understand with data.

## Figures and Tables

**Figure 1 ijerph-20-03299-f001:**
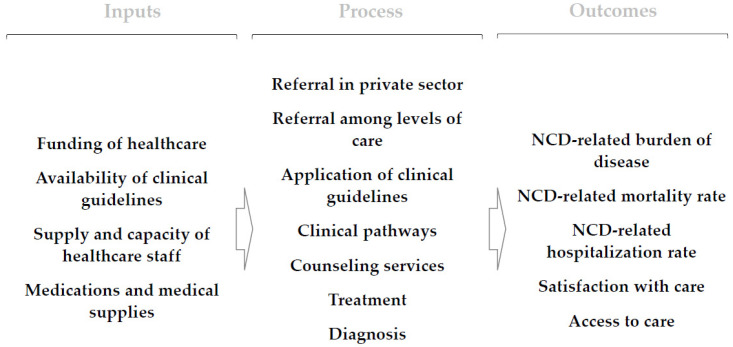
The conceptual framework for analysis of NCDs management in Kosovo.

**Figure 2 ijerph-20-03299-f002:**
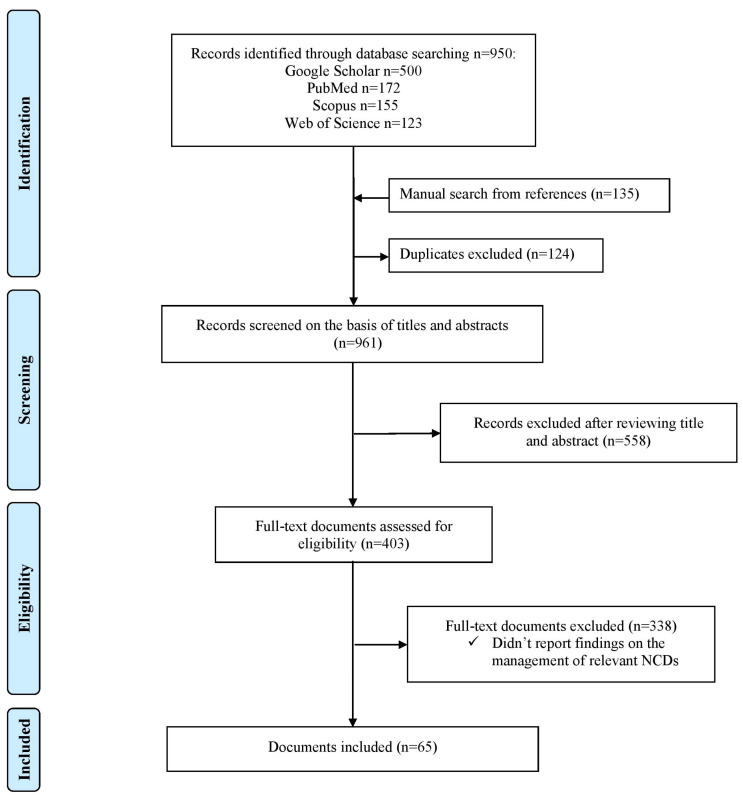
Selection of studies.

**Table 1 ijerph-20-03299-t001:** Study characteristics.

Author	Year	Type of Publication	Factors Influencing the Management of NCDs (Subthemes Reported)	Information on the Management of NCDs (Subthemes Reported)	Outcomes of Management of NCDs (Subthemes Reported)	NCDs Reported
Barbullushi et al. [21]	2002	Journal article		Treatment, clinical pathways, application of clinical guidelines		Overall
Haxhikadrija et al. [22]	2008	Journal article			NCD-related mortality rate	Breast cancer
Knowles et al. [23]	2008	Report	Supply and capacity of healthcare staff	Diagnosis, treatment, clinical pathways, application of clinical guidelines	NCD-related burden of disease	Breast cancer, cervical cancer
Schneider et al. [24]	2008	Report	Funding of healthcare	Referral in private sector		Overall
Ajvazi et al. [25]	2009	Journal article	Availability of clinical guidelines			Diabetes
Balkan Investigative Reporting Network [26]	2009	Report	Medications and medical supplies	Referral in private sector		Overall
Devolli-Disha et al. [27]	2009	Journal article		Diagnosis		Breast cancer
FRIDOM [28]	2009	Report	Medications and medical supplies, availability of clinical guidelines	Referral in private sector		Overall
Burkle [29]	2010	Journal article			NCD-related mortality rate	Overall
Percival et al. [30]	2010	Journal article			NCD-related mortality rate	Overall
Bakalli et al. [31]	2011	Journal article			NCD-related mortality rate	Hypertension
Begolli et al. [32]	2011	Report		Diagnosis, referral among levels of care, referral in private sector		Overall
Bielecka-Dabrowa et al. [33]	2011	Journal article		Diagnosis, treatment, application of clinical guidelines		Hypertension
Daullxhiu et al. [34]	2011	Journal article			NCD-related mortality rate	Hypertension
Health for All [35]	2011	Report			Access to care	Overall
O’Hanlon et al. [36]	2011	Journal article	Supply and capacity of healthcare staff			Overall
Balkan Investigative Reporting Network [37]	2012	Report	Medications and medical supplies	Referral among levels of care		Overall, diabetes
Luta et al. [38]	2012	Journal article			Access to care	Overall
Ministry of Health [39]	2012	Report		Diagnosis, treatment		Overall
Arifi et al. [40]	2013	Journal article			NCD-related mortality rate	Diabetes
Hee Lee-Kwan et al. [41]	2013	Report	Funding of healthcare	Referral in private sector		Overall
Hoxha [42]	2013	Report	Medications and medical supplies, supply and capacity of healthcare staff, funding of healthcare	Treatment, counseling services	Satisfaction with care	Overall, diabetes, breast cancer, cervical cancer
Balidemaj et al. [43]	2014	Journal article	Supply and capacity of healthcare staff	Treatment		Overall
Bhabha et al. [44]	2014	Report			Access to care	Overall
Tahiri et al. [45]	2014	Journal article			Satisfaction with care	Overall
Uka [46]	2014	Report	Funding of healthcare			Overall
Vian [47]	2014	Report	Medications and medical supplies, funding of healthcare			Overall, breast cancer, cervical cancer
Davies et al. [48]	2015	Report	Supply and capacity of healthcare staff, availability of clinical guidelines	Diagnosis, treatment, clinical pathways, application of clinical guidelines		Breast cancer, cervical cancer
Hoxha et al. [49]	2015	Policy brief	Medications and medical supplies, funding of healthcare	Treatment, counseling services, referral in private sector		Diabetes
Hoxha et al. [50]	2015	Policy brief		Treatment	Access to care, satisfaction with care	Diabetes
Hoxha et al. [51]	2015	Report	Medications and medical supplies, funding of healthcare	Referral in private sector		Overall
Raunio et al. [52]	2015	Policy brief	Funding of healthcare		Access to care	Overall
Raunio et al. [53]	2015	Policy brief			Access to care	Overall
Dixit et al. [54]	2016	Report		Counseling services	Access to care	Breast cancer
Farnsworth et al. [55]	2016	Report		Diagnosis	Access to care	Hypertension, breast cancer
Giordano et al. [56]	2016	Journal article		Diagnosis		Breast cancer
Kantar TNS [57]	2016	Report	Availability of clinical guidelines	Treatment, referral in private sector		Diabetes, hypertension
Moore et al. [58]	2016	Journal article	Availability of clinical guidelines			Overall
Thompson et al. [59]	2016	Journal article	Availability of clinical guidelines	Counseling services		Hypertension
Ramadani et al. [60]	2016	Journal article			NCD-related mortality rate	Other
Zahorka et al. [61]	2016	Report		Diagnosis, treatment		Hypertension
Zahorka et al. [62]	2016	Report		Referral among levels of care		Overall
Hoxha et al. [63]	2017	Policy brief		Referral among levels of care		Overall
Hoxha [64]	2017	Report		Referral in private sector		Overall
Hoxha et al. [65]	2017	Policy brief	Funding of healthcare			Overall, diabetes
Hoxha et al. [66]	2017	Policy brief			Satisfaction with care	Overall
Hughes et al. [67]	2017	Journal article		Treatment, clinical pathways		Diabetes
Bajraktari et al. [68]	2018	Journal article		Treatment	NCD-related mortality rate	Hypertension
Jakupi et al. [69]	2018	Journal article	Medications and medical supplies, funding of healthcare			Other
Zejnullahu-Raci et al. [70]	2018	Journal article		Diagnosis, treatment	NCD-related mortality rate	Cervical cancer
Bytyci et al. [71]	2019	Journal article		Counseling services, clinical pathways		Diabetes
Bytyqi-Damoni et al. [72]	2019	Journal article		Treatment		Diabetes
Cuperjani et al. [73]	2019	Journal article		Treatment		Breast cancer
World Health Organization [74]	2019	Report	Availability of clinical guidelines	Application of clinical guidelines	NCD-related hospitalization rate	Diabetes, hypertension
Ymerhalili et al. [75]	2019	Journal article		Clinical pathways		Diabetes, hypertension
Dimitrova et al. [76]	2020	Journal article	Availability of clinical guidelines			Overall
Eyvazzadeh et al. [77]	2021	Report		Referral among levels of care		Overall, breast cancer
Milosevic et al. [78]	2021	Journal article			Satisfaction with care	Overall
Podvorica et al. [79]	2021	Journal article		Counseling services		Other
Bytyci Katanolli et al. [80]	2022	Journal article		Counseling services		Overall
Bytyci Katanolli et al. [81]	2022	Journal article		Counseling services		Overall
Ejupi et al. [82]	2022	Journal article		Counseling services, treatment		Breast cancer
Hoxha et al. [83]	2022	Journal article		Treatment		Breast cancer
Hoxha et al. [84]	2022	Journal article	Application of clinical guidelines			Other
Obas et al. [85]	2022	Journal article		Diagnosis, treatment		Diabetes, hypertension, other

NCD = Non Communicable Disease, UCCK = University Clinical Center of Kosovo

## Data Availability

No data available.

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
