# Peer review of "Management of Non-Communicable Diseases in Kosovo: A Scoping Review"

_ijerph, 2023, doi:10.3390/ijerph20043299_

Round 1
Reviewer 1 Report
Dear author team,
Interesting publication you have submitted.
With much interest, I did read your manuscript and overall, I think your contribution is highly relevant.
I have some queries related to subchapter seeking for clarification, additions, and maybe revisions. Overall, the weakest chapter in my opinion is the methods chapter.
Methods:
As a reader I have difficulties to exactly understand the search strategy. Its not clear, when exactly (date) you did apply your search, as that determines also, in how far its replicable. But is has to be replicable and others need to be able to exactly find the same sources you have included in your presentation. Some of the reports/policy briefs don’t have the mention of NCDs in the title, neither in the abstract. Other publications have not been included in your selection, but carry NCDs in their title and abstract for e.g. see the following. Pls. clarify why these sources have not been considered.
· Bytyci-Katanolli, Ariana, et al. "Non-communicable disease prevention in Kosovo: quantitative and qualitative assessment of uptake and barriers of an intervention for healthier lifestyles in primary healthcare." BMC health services research 22.1 (2022): 1-17.
· Bytyci Katanolli, Ariana, et al. "Perceived barriers to physical activity behaviour among patients with diabetes and hypertension in Kosovo: a qualitative study." BMC Primary Care 23.1 (2022): 257.
· Obas, Katrina Ann, et al. "Study protocol: a prospective cohort on non-communicable diseases among primary healthcare users living in Kosovo (KOSCO)." BMJ open 10.9 (2020): e038889.
· Obas, Katrina. "Kosovo Non-Communicable Disease Cohort (KOSCO): baseline results of a prospective primary healthcare user-based longitudinal study on the prevention and control of chronic diseases." International Journal of Integrated Care 21.S1 (2021).
Framework:
1. The focus of the paper on “clinical treatment/management” of NCDs is surprising. A public health perspective on NCDs should consider social determinants and risk factors for NCDs and have a strong focus on prevention measures. Suggest to either expand the framework and include more aspects of the analyzed papers or state its limited focus as limitation of the paper.
Results:
As the results are presented now, there is no novelty in terms of content to it. The reader does not learn anything new about NCDs. Did you find any innovation presented in any of the papers? What is unique about NCD prevention and management in Kosovo?
2. An overview table of all articles you have analyzed should be presented in the results section and not only in the supplement. You can build on the table you have, but organize it alphabetically (so the reader can see who are the major author teams you are presenting) and try to present the most interesting meta narrative you have founds across all these publications.
Discussion:
3. A critical discussion towards state of the art NCD publications/technical reports would benefit your presentation of novelty in this subchapter.
References:
4. Just browsing the literature list, I realized some few inconsistencies
Author Response
See attached document.

Reviewer 2 Report
Generally, this is a very well-written manuscript addressing an important public health topic. The paper is well-organized and contains all of the components and sections which are very well-developed. The methodology is clearly explained and the paper presents a sound synthesis of the literature addressing the topic.
Some points for consideration for the benefit of the paper include the following:
1. Methods: Information sources, search strategy, and eligibility criteria (page 3): under this section, the authors should consider mentioning and explaining more clearly the search strategy terms related to NCDs and terms referring to the country Kosovo and not only referring it to the Online Appendix. It would be good to explain why the alternative term “sëmundjet kronike” used for Google Scholar (Albanian language search) was not used in English language search (“chronic conditions” or “chronic diseases”).
2. A few errors will have to be corrected before publication: i.e. Under Results/ Characteristics of sources of evidence (page 5): “... The documents were published between 2022 to 2022.” It might be “.. between 2010 to 2022”.
Results/ Management of NCDs (page 7): “... From the initial presentation to definitive therapy, the patient’s challenging and ineffective.” The sentence should be completed.
Author Response
See attached document.
